# Could MMP3 and MMP9 Serve as Biomarkers in EBV-Related Oropharyngeal Cancer

**DOI:** 10.3390/ijms25052561

**Published:** 2024-02-22

**Authors:** Anna Polz, Kamal Morshed, Bartłomiej Drop, Małgorzata Polz-Dacewicz

**Affiliations:** 1Genomed S.A., 02-971 Warsaw, Poland; anna.polz@genomed.pl; 2Department of Otolaryngology Head and Neck Cancer, University of Technology and Humanities in Radom, 26-600 Radom, Poland; k.morshed@uthrad.pl; 3Department of Computer Science and Medical Statistics with e-health Laboratory, Medical University of Lublin, 20-090 Lublin, Poland; bartlomiej.drop@umlub.pl; 4Department of Virology with Viral Diagnostics Laboratory, Medical University of Lublin, 20-093 Lublin, Poland

**Keywords:** MMP3, MMP 9, EBV, oropharyngeal cancer, biomarkers

## Abstract

The high incidence of, and mortality from, head and neck cancers (HNCs), including those related to Epstein–Barr virus (EBV), constitute a major challenge for modern medicine, both in terms of diagnosis and treatment. Therefore, many researchers have made efforts to identify diagnostic and prognostic factors. The aim of this study was to evaluate the diagnostic usefulness of matrix metalloproteinase 3 (MMP 3) and matrix metalloproteinase 9 (MMP 9) in EBV positive oropharyngeal squamous cell carcinoma (OPSCC) patients. For this purpose, the level of these MMPs in the serum of patients with EBV-positive OPSCC was analyzed in relation to the degree of histological differentiation and TNM classification. Our research team’s results indicate that the level of both MMPs is much higher in the EBV positive OPSCC patients compared to the EBV negative and control groups. Moreover, their levels were higher in more advanced clinical stages. Considering the possible correlation between the level of MMP 3, MMP 9 and anti-EBV antibodies, and also viral load, after statistical analysis using multiple linear regression, their high correlation was demonstrated. The obtained results confirm the diagnostic accuracy for MMP 3 and MMP 9. Both MMPs may be useful in the diagnosis of EBV positive OPSCC patients.

## 1. Introduction

Cancer is a serious public health problem and one of the most important challenges to modern medicine. In 2022, the International Agency for Research on Cancer recorded 20.0 million new cancer cases and almost 10.0 million cancer deaths worldwide. Head and neck cancer (HNC) is the seventh most common cancer all over the world. More than 890,000 new cases and 450,000 deaths are reported annually [1]. The incidence includes approximately 380,000 cases of cancer of the lip and oral cavity, 185,000 of the larynx, 133,000 of the nasopharynx, 98,000 of the oropharynx, 84,000 of the hypopharynx, and 54,000 of the salivary glands. It is estimated that in 2045 the number of new cancer cases will reach 32.6 million cases.

In Poland, approximately 5000 new cases of HNC are registered every year [2]. In 2020, as indicated by the Globocan registry data, 1659 new cases of oropharyngeal cancer were registered and only 297 cases of nasopharyngeal cancer (NPC) [3]. The most frequently diagnosed type is squamous cell carcinoma, which originates from the epithelium of the oral cavity, pharynx and larynx [4]. 

The etiopathogenesis of HNC, including OPSCC, is multifactorial. In addition to the well-documented role of environmental and lifestyle factors, oncogenic viruses, mainly HPV but also EBV, play a role in the development of oropharyngeal cancer [5].

Pathological evaluation of surgical specimens should include tumor size, growth pattern, depth of invasion (DOI) for oral cancer, total number of lymph nodes removed, number of involved lymph nodes and their location, presence of extracapsular lymph node dilatation, perineural status, and lymphatic infiltration and surgical margins [6]. The above-mentioned features are important for assessing the severity of the disease and prognosis, as well as for determining postoperative adjuvant treatment. All patients with newly diagnosed oropharyngeal SCC should undergo HPV evaluation using p16 immunohistochemistry (IHC).

It should be emphasized that there has been significant progress in the treatment of diseases caused by the HPV virus [7]. In addition, patients with human papillomavirus- (HPV-) related oropharyngeal squamous cell carcinoma (OPSCC) have a better prognosis than for HPV-negative OPSCC when treated with standard chemoradiotherapy. Among HPV negative OPSCC cases, some are caused by persistent EBV infection.

Epstein–Barr virus (EBV), also called human herpes virus 4 (HHV-4), has been classified in the *Orthoherpesviridae* family, subfamily *Gammaherpesvirinae*, genus *Lymphocryptovirus* [8]. Approximately 1.5% of all human malignancies and 1.8% of cancer deaths are related to EBV infection [9]. These include Burkitt’s lymphoma, Hodgkin’s lymphoma, B cells and T cells, and natural killer cell lymphoma, gastric cancer (GC), and nasopharyngeal cancer (NPC). NPC and GC account for as many as 82% of cases and 89% of deaths from EBV-related cancers.

In infected cells, EBV establishes a latency period which, under the influence of various factors, may periodically reactivate to the lytic phase [10]. More and more scientific evidence indicates that the proteins of the lytic phase of the EBV virus play a pivotal role in oncogenesis [11,12]. Most of the research available in the global medical literature concerns NPC, the most common cancer in the Asian population [13]. However, there are few studies focusing on another anatomical location, i.e., oropharyngeal cancer. 

Matrix metalloproteinases (MMPs) belong to a family of zinc-dependent endopeptidases that have the ability to degrade various extracellular matrix (ECM) proteins. In the complex network of macromolecules, the dominant protein is collagen, which is very resistant to the action of proteinases. The only enzymes that are able to degrade collagen are MMPs [14]. The first MMP was discovered in 1962 by Gross and Lapiere [15] during studies on tadpole tail resorption. 

All MMPs have a similar basic molecular structure [16]. Specifically, they contain three conserved common domains: a pro-peptide domain, a catalytic domain containing a zinc ion-binding motif, and a hemopexin-like domain at the C-terminus, connected to the catalytic domain. Human MMPs consist of 26 members and are classified as secreted and membrane-anchored MMPs based on their structural features. Taking into account their specificity towards ECM components, secreted MMPs are further divided into collagenases, gelatinases, stromelysins, matrilysins and other groups. 

Under normal conditions, MMPs are responsible for tissue homeostasis by maintaining the complex structure of the ECM and its properties. They play a key role in various physiological processes, such as remodelling, embryonic development, wound healing, but also in pathological ones, e.g., inflammation, carcinogenesis and migration. MMPs are capable of degrading many components in ECM or basement membrane (BM), initiating and promoting blood vessel formation which is crucial for malignant tumor development and progression [17,18]. MMPs may have tumor-suppressive and tumor-promoting effect depending on the cancer type or on a specific tissue where malignancy is developing. They were also found to be involved in tumor metastasis [19]. 

The main biological role of MMPs is the degradation of ECM proteins and glycoproteins, membrane receptors, cytokines and growth factors [17,18,19]. Deregulation of MMP activity leads to the development of various pathologies, which can be divided into tissue destruction, fibrosis and matrix weakening. 

MMP expression is tightly controlled in terms of transcription, secretion, activation and inhibition of the activated enzyme. Cancer progression and metastasis depend on the proteolytic activity of numerous matrix metalloproteinases (MMPs). They determine tissue integrity and the expression of immune cells. The proteolytic activity of MMPs depends on gene expression, mRNA stability, compartmentalization in vesicles and membrane microdomains in the case of membrane-bound MMPs, and activation from the inactive zymogen form and inhibition of proteolysis. 

MMP gene expression is regulated, among others, through the NF-κB, MAPK and JAK/STAT signaling pathway via cell–matrix and cell–cell interactions, as well as growth factors, glucocorticoids, cytokines, retinoic acid, and interleukins. Some MMPs are not expressed in cells, but their expression is induced by exogenous signals, such as cytokines, growth factors, hormones, and changes in cell–matrix and cell–cell interactions. Moreover, MMP activation can also occur via physiochemical agents, such as heat, low pH, and reactive oxygen species. The proteolytic activity of MMPs is tightly controlled by the tissue inhibitors of metalloproteinases (TIMPs). So far, four TIMPs have been discovered, respectively TIMP-1 to TIMP-4, and their expression is controlled during tissue development and differentiation. Often, in pathological conditions in which an increase in MMP activity is observed, the level of TIMPs directly responsible for MMP activity decreases at the same time. 

MMPs are secreted by many cells, such as fibroblasts, vascular smooth muscle (VSM), and leukocyte macrophages, neutrophils, and lymphocytes. Regulation of MMPs occurs at the level of mRNA expression and by activation of their latent zymogen form. Furthermore, MMPs can be secreted in the inactive form proMMP, which is cleaved to the active form by various proteinases, including other MMPs [20]. Many MMPs have been shown to be overexpressed in head and neck cancer: MMP-1, MMP-2, MMP-3, MMP-7, MMP-8, MMP-9, MMP-10, MMP-11, MMP-13, and MMP-14 [21]. 

MMP3 is also called stromelysin-1 as a representative of the stromelysin subfamily and degrades, among others, stromelysin 1, collagen types III, IV, IX, X, fibronectin, and laminin [22,23]. It also participates in the breakdown of the tight junctions mediated by E-cadherin, which results in losing contact between tumor cells and the surrounding cells, thus promoting the invasion capacity of the tumor cells. Moreover, it promotes the epithelial–mesenchymal transition, a process connected with changes in the epithelial cells, which allows their migration through the basement membrane. 

MMP 9. also known as gelatinases B, degrades gelatine type I and V, collagen type IV and V and fibronectin [24]. A special interest regarding tumorigenesis is paid to MMP 9, since collagen IV, which may be selectively degraded by MMP 9, is present in BM. MMP 9 is expressed by endothelial cells, osteoclasts, chondrocytes, osteoblasts, and malignant cells. 

Many studies have revealed increased level of MMPs in human cancers, including head and neck cancer [22,23,24]. Among many metalloproteinases, MMP3 and MMP 9 have attracted particular attention from researchers due to their overexpression in EBV-related NPC [25,26]. Both may be potential diagnostic or prognostic biomarkers for certain types of cancer, including head and neck squamous cell carcinoma. We were inspired to choose these MMPs by two publications in particular. The first is by Li et al. [25], in which the authors suggests that the combination of MMP 3 activity and EBV antibodies may be a useful biomarker in the diagnosis of NPC. In turn, in the second publication, Lan et al. [26] discovered that, in addition to MMP 9, the new target gene is MMP 3, the expression of which is increased by Zta—a lytic trans-activator of Epstein-Barr virus (EBV). Both researchers focused on NPC. However, we wanted to test whether these MMPs might play a similar role in EBV-related oropharyngeal cancer. 

Therefore, in the current study, we aimed to investigate the serum levels of MMP 3 and MMP 9 in patients with EBV-positive and EBV-negative OPSCC compared to the control group. We tried to check whether these MMPs could serve as diagnostic and/or prognostic biomarkers in EBV positive OPSCC. For this purpose, the relationship between the level of these MMPs, histological differentiation (grading) and TN classification was analyzed. Moreover, in the analysis, we took into account a possible correlation between the level of tested MMPs and the level of various types of anti-EBV antibodies, as well as viral load. We also assessed the accuracy of the tested MMPs. 

## 2. Results

### 2.1. Evaluation of Serum Level of MMP 3 and MMP 9 in EBV Positive and EBV Negative Oropharyngeal Cancer Patients in Comparison to the Control Group

In the first stage of our research, an ELISA test was carried out to determine the level of MMP 3 and MMP 9 in the serum of the examined individuals. A comparison was made between groups of patients with EBV positive oropharyngeal cancer, EBV negative, and those in whom any cancer was excluded, as graphically presented in Figure 1. Both MMP 3 (Figure 1a) and MMP 9 (Figure 1b) levels were significantly higher among EBV-positive OPSCC patients (*p* < 0.0001). The exact values of the tested parameters are listed in Table 1.

### 2.2. Evaluation of Serum Level of MMP 3 and MMP 9 by Grading (G) and T, N Classification among EBV Positive Oropharyngeal Cancer Patients

Then, we assessed the concentration of both metalloproteinases depending on the degree of tumor differentiation and T, N classification. The highest concentrations of MMP 3 (Figure 2a) and MMP 9 (Figure 2d) were observed in poorly differentiated tumors (G3). This difference was highly statistically significant (*p* < 0.0001). Due to small numbers, T and N features were analyzed in the following subgroups: T1−T2, T3−T4, and N0−N1, N2−N3. As shown in the data presented in Figure 2, the concentration of both MMP3 (Figure 2b,c) and MMP 9 (Figure 2e,f) depended on the clinical stage and was highest in more advanced stages, i.e., T3−T4 and N2−N3 (*p* < 0.0001).

### 2.3. Correlation between the Serum Levels of Tested MMPs, All Types of Anti-EBV Antibodies and Viral Load in EBV-Positive OPSCC Patients

EBV DNA was detected in tumor tissue of OPSCC patients. We sought to semi-quantitatively define EBV viral load as low or high based on the cycle threshold (Ct) value of the viral gene. The result was considered high when the Ct value of the viral gene was <38 and low when the Ct value was ≤38. Table 2 shows percentage of tested samples showing low or high viral load according to grading. In all G1 cases, the viral load was low, while in G3 it was high (*p* < 0.0001). 

Similar differences in the viral load concerned T and N stages (Table 3). In more advanced clinical stages, high viral load was demonstrated in both T3−T4 (75.0%) and N2−N3 (87.5%).

### 2.4. Correlation between the Serum Levels of Tested MMPs, All Types of Anti-EBV Antibodies and Viral Load in EBV-Positive OPSCC Patients

In turn, at this stage we wanted to demonstrate a possible correlation between the tested MMPs and anti-EBV antibodies and viral load (Figure 3). For this purpose, we used the results of our previous studies in which we analyzed different types of anti-EBV antibodies [27]. After statistical analysis using multiple linear regression, the high correlation between MMP 3 and MMP 9 and other tested parameters was demonstrated. 

The strong positive correlation was observed between: MMP 3 and EBNA1 IgA *p* < 0.0001; EBNA 1 IgG *p* < 0.0001; EBVCA IgA *p* < 0.0001;MMP 9 and EBNA1 IgA *p* < 0.0001, EBNA1 IgG *p* = 0.0001; EBVCA IgA *p* < 0.0001; EBVCA IgG *p* < 0.0001;MMP 3, MMP 9 and viral load *p* < 0.0001.

Moreover, the analysis showed the strong positive correlation between MMP3, MMP 9 and antibodies against main EBV oncoprotein—LMP 1 both in IgA and IgG classes *p* < 0.0001, as well as with anti-Zta antibodies both in IgA and IgG classes *p* < 0.0001. 

However, no correlation was found between the tested MMPs and anti-EA antibodies, both in the IgA and IgG classes, as well as anti-EBVCA IgG antibodies. 

### 2.5. Receiver Operating Characteristic (ROC) Curve Analysis to Determine the Diagnostic Accuracy of Serum MMP 3 and MMP 9 Level in OPSCC Patients EBV Positive vs. OPSCC Patients EBV Negative

In the final stage of our research, we tried to assess the accuracy of the tested MMPs, i.e., whether MMP 3 and MMP 9 can be good biomarkers in the diagnosis of patients with OPSCC. For this purpose, ROC curve analysis was used to compare the levels of both MMP 3 and MMP 9 in the serum of EBV-positive OPSCC patients with the group of EBV-negative subjects (Figure 4). As shown by the area under the curve (AUC), the MMP 3 level (Figure 4a) was found to be a sensitive and specific parameter in identifying patients with EBV-positive OPSCC (AUC = 0.9297; Std. Error = 0.0221; 95% CI 0.8863–1.000; *p* < 0.0001). Similar results were obtained for MMP 9 (Figure 4b) (AUC 0.9481; Std. Error = 0.0291; 95% CI 0.8909–1.000; *p* < 0.0001). 

## 3. Discussion

Extensive molecular studies of various malignancies, including HNC, have shown that the process of carcinogenesis is influenced by genetic and epigenetic changes in cancer cells, as well as rearrangement of components of the tumor microenvironment (TME) [24]. Many changes in the TME are mediated by MMPs, which are produced not only by malignant epithelial cells but also by stromal cells, such as fibroblasts and endothelial cells [18,21,24]. MMPs are responsible for tissue homeostasis. Therefore, dysregulation of MMP expression may lead to the development of many diseases, such as atherosclerosis, osteoarthritis, periodontal disease, respiratory tract disorders, glomerulonephritis, inflammatory bowel disease, neurodegeneration, chronic obstructive pulmonary disease, multiple sclerosis, and cardiovascular disease. In addition, they play a role in the development and progression of cancers [28]. 

Current scientific evidence shows that at least 40% of all cancer cases could be prevented with effective primary prevention, and further mortality could be reduced by early detection of cancer.

It has long been known that overexpression of MMPs influences the risk of development and prognosis of various cancers. Nevertheless, the utility of MMPs as biomarkers is relatively new. The ideal biomarker should be easily measurable, accurate, non-invasive, sensitive, inexpensive and easy to perform [29]. Therefore, many researchers evaluate the level of MMPs depending on the clinical stage of the cancer as a potential diagnostic and prognostic marker in HNSCC [22,23,24,25].

Numerous studies have shown increased MMP activity in head and neck cancers of various locations, i.e., hypopharynx, nasopharynx, larynx and oral squamous cell carcinoma [17,19,21,22,30,31,32,33]. Some researchers have analyzed the histological expression of MMPs in tumor tissue, while others have examined the concentration of MMPs in the serum of patients with head and neck cancer. In our study, we analyzed the level of MMPs in the serum of patients with OPSCC.

Both MMP3 and MMP 9 play a pivotal role in the degradation of the extracellular matrix causing tumor invasion, metastasis and vascularization of tumor tissue [34,35].

Increased concentration of MMP9 in serum compared to healthy people was observed by Lotfi et al. [36], while Tadbir et al. et al. [37] found increased MMP3 concentration in the serum of OSCC patients, suggesting that MMP3 may be a helpful diagnostic marker. The results of studies conducted by many other authors indicate high levels of MMP 3 and MMP9 as well as a positive correlation with the clinical stage in patients with NPC [38,39,40].

Our own research showed significantly higher levels of both MMP3 and MMP 9 in the serum of patients with EBV positive oropharyngeal cancer compared to EBV negative patients and the control group. As mentioned in the introduction, the research presented in the available medical literature basically concerns NPC patients. Although we analysed a different tumor location, i.e., oropharyngeal cancer, our results are similar.

Analysing the level of tested MMPs from the clinical stage, we showed a significant relationship. We observed that the concentration of both tested MMPs in serum depended on the degree of tumor differentiation and T, N classification. The highest concentrations of both MMP3 and MMP 9 were observed in poorly differentiated tumors (G3). They were also the highest in more advanced clinical stages, i.e., T3–T4 and N2–N3. By semi-quantitative analysis of EBV load, we observed higher viral load in G3 as well as in T3–T4 and in N2–N3 stages.

EBV infection plays an essential role in the development and progression of nasopharyngeal carcinoma. Late diagnosis is one of the reasons for high mortality from HNCs. Therefore, early diagnosis is extremely important. Many studies suggest that serological assessment of anti-EBV antibody levels is an effective tool in the early detection of NPC [41]. Traditional screening tests are based on serology, including EBV IgA markers for viral capsid antigen (VCA), early antigen (EA), and EBV nuclear antigen 1 (EBNA 1). Determining EBV DNA load also plays an important role in NPC screening tests [42,43,44].

Biomarkers differ in specificity and sensitivity, hence currently using a combination of biomarkers is one of the diagnostic strategies [45]. Combined testing of EBV antibodies and EBV DNA in plasma by PCR is recommended to predict prognosis and stratify treatment [9]. As shown by Simon et al. [46], multiplex serology is well suited for comprehensive assessment of antigens and antibodies but is not used in large-scale studies. 

Our team’s research is part of this direction in searching for diagnostic and/or prognostic biomarkers in HNCs, with particular emphasis on EBV-related oropharyngeal cancer. We are conducting research on various aspects of EBV-related oropharyngeal cancer, testing various molecules in serum, saliva and tumor tissue as potential biomarkers with practical applications.

In our recent study, we examined the serum prevalence and the level of anti-Zta and anti-LMP1 antibodies [27]. We concluded that combined antibody testing should be performed to increase diagnostic accuracy.

In the same group of patients mentioned above, the levels of MMP 3 and MMP 9 were analyzed to determine their usefulness as diagnostic and prognostic markers in EBV positive OPSCC patients. For this purpose, using multiple regression analysis, we checked the possible correlation between the levels of MMP 3, MMP 9, the level of different types of anti-EBV antibodies, and viral load. We found a strong positive correlation of both MMP 3 and MMP 9 with anti- EBNA 1 IgA, IgG, EBVCA IgA antibodies, and viral load. However, no correlation was found between the tested MMPs and anti-EA antibodies. 

Li et al. [25] presented interesting results of a study in which they assessed the diagnostic value of MMP 3 in combination with anti-EA IgA and anti-VCA IgA antibodies. These authors demonstrated that MMP 3 activity is a better marker than serum MMP 3 protein concentration. Moreover, they suggested that the cumulative assessment of MMP 3 activity and EBV antibodies increased potential diagnostic value for NPC. 

EBV, like other herpes viruses, has the ability to establish latency, which may periodically switch into the lytic phase. The first protein in the lytic phase of EBV is the Z protein, a product of the BZLF1 gene, also called Zta or ZEBRA [47,48]. Lan et al. [26], analyzing the expression profile and biological function of Zta-induced MMP 3 and MMP 9, demonstrated that both MMPs are induced by Zta and only MMP3 is required for Zta-induced cell migration. 

Zhang et al. [49] conducted a systematic assessment of the diagnostic value of Zta antibodies in the serum of patients with NPC, demonstrating their high diagnostic value. 

In the present study, we demonstrated the strong positive correlation between the level of MMP3, MMP 9 and the level of anti-Zta antibodies both in IgA and IgG classes. 

Moreover, the analysis showed the strong positive correlation between MMP3, MMP 9 and antibodies against main EBV oncoprotein LMP 1, both in IgA and IgG classes. 

LMP 1, the main oncoprotein of EBV, has the ability to modulate the expression or activity of various oncogenes [10,11,12,13]. LMP 1 induces the expression of pro-inflammatory cytokines, protecting cells from apoptosis. Furthermore, LMP 1 modulates cell–matrix interactions by inducing matrix metalloproteinases (MMPs). LMP 1 increased the expression of MMP 3 and MMP 9 in NPC [50,51]. 

LMP 1 regulates multiple signaling pathways, including NF-κB, influencing cell proliferation, apoptosis, transformation, metastasis, and invasion [52]. Moreover, by increasing the susceptibility of cells to the virus through the secretion of MMPs, it facilitates the degradation of the extracellular matrix. LMP 1 interferes also with the stability of tumor suppressor gene p53, inhibiting apoptosis.

By producing Zta and LMP 1, EBV increases the expression of MMPs, which may disrupt the continuity of cell basement membranes, which in turn facilitates the development of infection. In the process of malignant transformation, cancer cells use virus-induced MMPs to develop tumors, increasing the rate of their proliferation, blood vessel formation and metastasis [26]. 

In the last stage of our study, using Receiver Operating Characteristic (ROC) Curve Analysis, we determined the diagnostic accuracy of serum MMP 3 and MMP 9 level in OPSCC patients EBV positive vs. OPSCC patients, and EBV negative. As shown by the area under the curve (AUC), both MMP 3 and MMP 9 levels were found to be a sensitive and specific parameters to identify patients with EBV-positive OPSCC. 

EBV infection plays an essential role in the development and progression of nasopharyngeal carcinoma. Late diagnosis is one of the reasons for high mortality from HNCs. Therefore, early diagnosis is extremely important. Many studies suggest that serological assessment of anti-EBV antibody levels is an effective tool in the early detection of NPC [41]. A very extensive meta-analysis on the diagnostic value of EBV DNA, EA IgA, VCA IgA, EBNA1 IgA, and Rta IgG, including 8382 patients with NPC and 15,089 controls, was presented by Liu et al. [53]. These authors concluded that the analyzed parameters have high accuracy in early diagnosis of NPC. 

To our knowledge, this is the first study in a Polish population to determine the utility and the accuracy of MMP 3, MMP 9 in EBV-related OPSCC patients. 

As mentioned in the Introduction, the etiopathogenesis of HNC, including OPSCC, is multifactorial. In addition to the well-documented role of environmental and lifestyle factors (the use of tobacco products and/or alcohol consumption), oncogenic viruses play a role in the development of oropharyngeal cancer. In a broader aspect, the microbiome plays a role in oncogenesis [54]. 

A microbiota, a complex ecosystem of microorganisms consisting of bacteria, viruses, protozoa and fungi living in different niches of the human body, including the oral cavity, plays a key role in many metabolic functions. Modifications in the microbiota composition can lead to several diseases, including cancers. The impact of microflora on anticancer immunity depends on its composition, its relationship with cancer and the stage of cancer advancement [55]. Tumoral microbiota can regulate tumor cell physiology and immune response through various signaling pathways, such as ROS, β-catenin, TLR, ERK, NF-κB, and STING. 

TME is a complex, heterogeneous and constantly modified ecosystem [24]. Interactions between various molecules, as well as MMPs, promote the growth and invasion of cancer. Moreover, the interaction between host cells and viral agents can lead to the creation of a microenvironment conducive to oncogenesis [18,19,20,21,22,23,24]. 

EBV can integrate its genome into the host cell genome, establishes latent infection in affected host cells and reactivates in the head and neck epithelium, influencing the pathogenesis of EBV-associated cancer [10]. Viral DNA stimulate the production of interferons and other cytokines, especially pro-inflammatory cytokines, which can modulate the immunological system of the host and contribute to the development of various diseases. 

Several studies have demonstrated an association between oral health and EBV infection, as well as a strong association of EBV with periodontitis [56]. Inflamed periodontal pockets may be a reservoir for the EBV virus, which will then infect oral epithelial cells. The salivary microbiome, circulating microbial DNA in blood, has been used as diagnostic biomarkers for many types of cancer, including OPSCC. It has been shown that microbiomes can promote or limit cancer development and progression by influencing tumor cells or the host immune system [54]. Microbes can also affect the effectiveness of cancer treatments, including radiotherapy, chemotherapy and immunotherapy. 

Taking into account the above-mentioned aspects, further research is necessary to clarify the role of EBV in the etiopathogenesis of OPSCC.

### Limitations

We are aware that our research is not without limitations. Firstly, the group of patients studied was relatively small. However, this is due to the fact that oropharyngeal cancer is not a common cancer in our region. Due also to the small number of subgroups T (tumor size) and N (lymph node involvement), they were analyzed as T1 − T2 and T3 − T4, and N0 − N1 and N2 − N3. 

Secondly, we did not detect EBV DNA in plasma. This was due to the fact that many different parameters were measured in the same clinical material and neither serum nor plasma was sufficient. This was also the reason why only two MMPs were selected instead of the entire panel. Many authors suggest that EBV DNA in plasma may be a useful diagnostic and prognostic biomarker [57]. This will be taken into account in subsequent specially planned studies on a new group of patients. Therefore, further research is needed to verify the observed trend. As Weixing Liu [53] wrote, “diagnostic accuracy is not affected by sample size or ethnicity. Given the small number of studies in non-Asian populations, the current results require confirmation in another population”. Therefore, despite these limitations, our research seems justified.

## 4. Materials and Methods 

### 4.1. Characteristics of Study Group

The study involved 110 patients with diagnosed and histo-pathologically confirmed squamous cell carcinoma of the oropharynx (OPSCC), hospitalized at the Department of Otolaryngology, Head and Neck Cancer, University of Technology and Humanities in Radom, Poland. The same group of patients was studied as in the previous publication [27]. The characteristics of the study groups are presented in Table 4.

#### Criteria Qualifying Patients for the Research Group

Patients for the study were selected based on the presence of EBV DNA in the tumor tissue—this was the basic qualifying criterion. The exclusion criterion was the presence of HPV DNA in the tumor tissue. For this reason, patients were assigned to a study group based on a negative result of the p16 immunohistochemical screening test, which was further verified using PCR. Only HPV negative patients were included in the study group. The US Joint Committee on Cancer TNM 8th edition recommends stratification pf all OPSCC cases by HPV status [58,59,60]. No patients had received radiotherapy or chemotherapy before.

The control group, matched in terms of sociodemographic features, consisted of 40 patients of the outpatient clinic in whom cancer was excluded. 

The research group (N = 110) included 58 patients in whom EBV DNA was detected in the tumor tissue, hereinafter referred to as EBV positive—EBV(+), and 52 patients without EBV DNA detected, hereinafter referred to as EBV negative—EBV(−). The age of the respondents ranged from 50 to 79 years. Therefore, two age groups were distinguished, i.e., 50–59 years and 60–79 years. The mean age of patients in the study group was 54.7 (SD = 2.6) and 68.5 (SD = 5.5), respectively. None of the EBV-positive or EBV-infected patients had distant metastases. Both groups did not differ significantly in terms of sociodemographic and clinical characteristics, so they had no impact on the analyzed parameters.

### 4.2. Clinical Specimens

Tissue and blood were collected from all cancer patients. Only blood was collected from the control group. HPV DNA and EBV DNA were detected in tumor tissue. However, MMP3, MMP9 were detected in the serum. Antibody results were taken from a previous study [27]. 

#### 4.2.1. Tissue Samples Collection

Tissue samples collected from all patients during the surgery were frozen at −80 °C and stored until analysis. During primary diagnosis, the classification of the tumor, node, and metastases (TNM) was determined according to the eighth edition of the TNM classification of head and neck cancer [55,56,57]. Histological grading was performed according to the World Health Organization criteria, which divide tumors into three types: well differentiated (G1), moderately differentiated (G2), and poorly differentiated (G3) [61].

#### 4.2.2. Serum Collection

Venous blood samples collected from all patients were centrifuged at 1500 rpm for 15 min at room temperature, and the sera were frozen at −80 °C until analysis. 

### 4.3. Molecular Methods

DNA Extraction from fresh-frozen tumor tissue and detection of EBV DNA and HPV DNA were performed as previously described [27]. 

### 4.4. Serological Methods

#### 4.4.1. MMP3 and MMP9

MMPs level in serum was determined by ELISA Kit for Matrix Metalloproteinase 3 (MMP3) (SEA101Hu) and ELISA Kit for Matrix Metalloproteinase 9 (MMP9) (SEA553Hu) according to the manufacturer’s recommendations (Cloud-Clone Corp., Katy, TX, USA). 

The minimum detectable dose in the MMP3 kit is usually less than 13.1 pg/mL; however. the minimum detection dose for MMP9 is usually less than 0.055 ng/mL. 

#### 4.4.2. Detection of EBV Antibodies

Serum antibody levels were determined using the commercially available Microblot–Array test (TestLine Clinical Diagnostics Ltd., Brno, Czech Republic) according to the manufacturer’s instructions. The Microblot–Array kits (CE IVD) were optimized and validated for the detection of IgA, IgG, and IgM antibodies in human serum, plasma, or cerebrospinal fluid. This test contained a combination of selected parts of the specific antigens of EBV (EBNA-1, EBNA-2, VCA p18, VCA p23, EA-D p54, EA-D p138, EA-R, Rta, ZEBRA, gp85, gp350, and LMP1).

### 4.5. Statistical Analysis

Tibco Statistica 13.3 (StatSoft, Kraków, Poland) and GraphPad Prism software version 10.1.1. (GraphPad, San Diego, CA, USA) were used to perform the data analyses. The Shapiro–Wilk test was used to test for a normal distribution of continuous variables. The relationship between clinical and demographic parameters was calculated using Pearson’s chi-square test. To compare differences between studied groups, the Mann–Whitney U test and/or Kruskal–Wallis Test were used. The correlation between all tested parameters was assessed using multiple linear regression analysis. Receiver Operating Characteristic (ROC) Curve Analysis was used to determine the diagnostic accuracy of serum MMP 3 and MMP 9 levels. 

## 5. Conclusions

The increasing morbidity and mortality from cancer in general, as well as HNC in particular, is a serious public health problem worldwide. This group includes cancers of various locations, including those in the oropharynx. Due to the fact that it is not a common cancer in our region, there are few studies on this location. Therefore, all these cases require new diagnostic methods and therapeutic strategies, as well as the search for targets for new anticancer drugs. The challenge of modern medicine is early diagnosis and effective treatment of cancer; hence, many researchers are making efforts to search for new and/or better biomarkers. 

To sum up, the obtained results demonstrated that both MMP3 and MMP9 levels are significantly higher in the group of EBV-positive OPSCC patients compared to EBV-negative subjects. The concentration of both MMPs was positively correlated with the progression of OPSCC and was significantly higher in more advanced stages. Furthermore, the concentration of both MMP 3 and MMP 9 was positively correlated with the level of anti-EBV antibodies, as well as EBV load. The analysis confirmed the diagnostic accuracy for both MMP 3 and MMP 9 in EBV positive OPSCC patients. 

The obtained results suggest that both MMP3 and MMP 9 in combination with anti-EBV antibodies may be valuable biomarkers in the diagnosis of EBV-positive OPSCC and may also be a useful tool for detecting and determining the stage of EBV infection.

We hope that our study will shed new insight into the usefulness of these non-invasive biomarkers in the diagnosis and prognosis of EBV-related OPSCC and will contribute to and encourage further research.

## Figures and Tables

**Figure 1 ijms-25-02561-f001:**
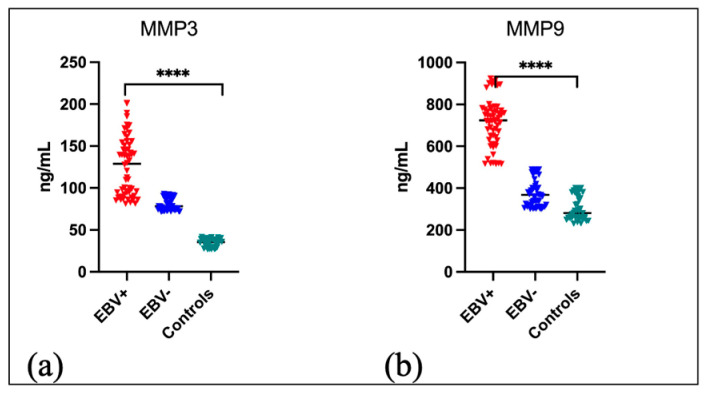
Serum level of MMP3 (**a**) and MMP9 (**b**) in EBV positive and EBV negative OPSCC patients in comparison to control group; Kruskal–Wallis Test; **** *p* < 0.0001.

**Figure 2 ijms-25-02561-f002:**
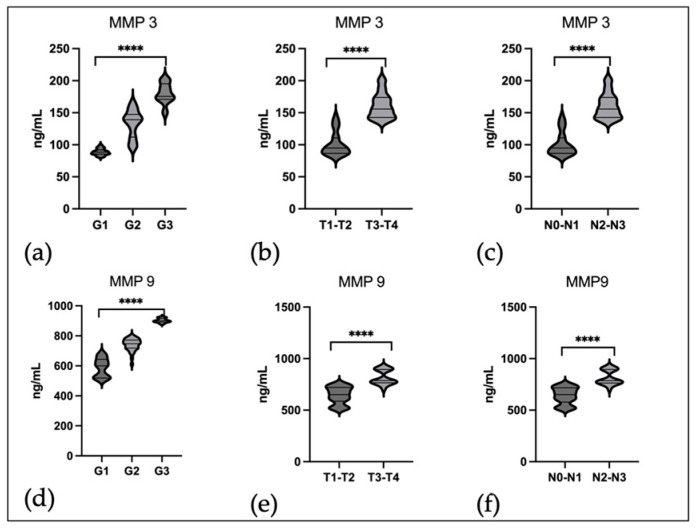
Serum level of MMP 3 and MMP 9 in EBV positive OPSCC: (**a**) serum level of MMP 3 in relation to grading; (**b**) serum level of MMP 3 in relation to T stages; (**c**) serum level of MMP 3 in relation to N stages; (**d**) serum level of MMP 9 in relation to grading; (**e**) serum level of MMP 9 in relation to T stages; (**f**) serum level of MMP 9 in relation to N stages. Kruskal–Wallis Test and Mann–White Test; **** *p* < 0.0001.

**Figure 3 ijms-25-02561-f003:**
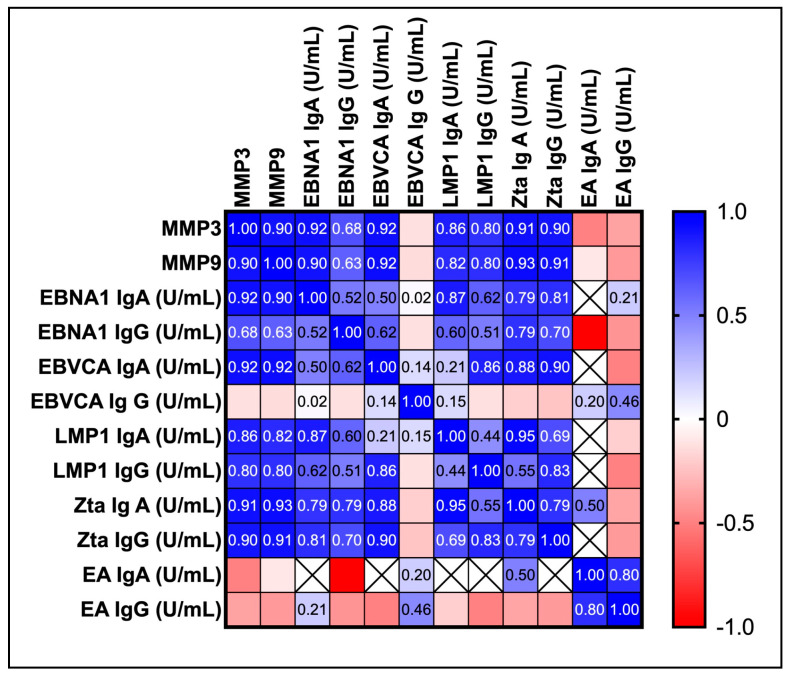
Correlation between the level of MMP3 and MMP9, anti-EBV antibodies and viral load in EBV positive OPSCC patients. Spearman’s rank coefficients are presented as the intensity of the colors. The closer Rs is to +1 or −1, the stronger the correlation. A perfect positive correlation is +1 (blue), and a perfect negative correlation is −1 (red).

**Figure 4 ijms-25-02561-f004:**
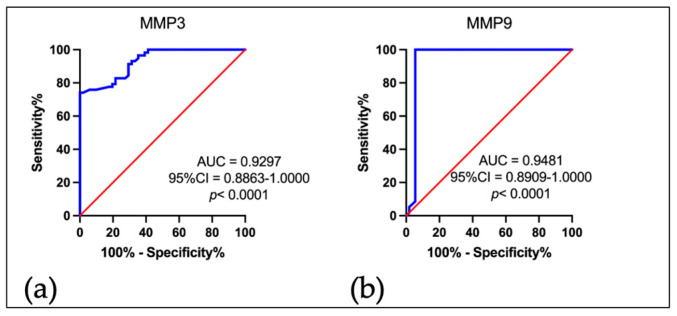
Receiver operating characteristic (ROC) analysis for MMP 3 (**a**) and MMP 9 (**b**). As shown by the area under the curve (AUC), the level of both MMPs was the most sensitive and specific parameter to determine EBV positive OPSCC patients. Blue line—serum level of tested MMPs (ng/mL).

**Table 1 ijms-25-02561-t001:** MMP 3 and MMP 9 concentration in serum EBV positive, EBV negative oropharyngeal cancer patients compared to control group.

MMP (ng/mL)	Group	Mean	Minimum	Maximum	SD	*p*-Value
MMP 3	EBV+	125.8	81.2	201.3	35.1	<0.0001 *
	EBV−	80.4	71.5	92.8	7.2	
	Control	34.8	26.8	41.5	5.0	
MMP 9	EBV+	715.7	515.5	923.4	115.3	<0.0001 *
	EBV−	379.2	300.8	489.6	65.7	
	Control	300.6	230.1	400.1	56.3	

* statistically significant; Kruskal–Wallis Test.

**Table 2 ijms-25-02561-t002:** EBV load in relation to grading (G) (%).

Viral Load	G1 N = 19	G2 N = 30	G3 N = 9	*p*-Value
Low	19 (100.0%)	18 (60.0%)	0	<0.0001 *
High	0	12 (40.0%)	9 (100.0%)	

* statistically significant.

**Table 3 ijms-25-02561-t003:** EBV load in relation to T,N classification (%).

**Viral** **Load**	**T1 − T2** **N = 34**	**T3 − T4** **N = 24**	* **p** * **-Value**
Low	7 (100.0%)	4 (25.0%)	0.0001 *
High	0	20 (75.0%)	
**Viral** **Load**	**N0 − N1** **N = 34**	**N2 − N3** **N = 24**	* **p** * **-Value**
Low	34 (100.0%)	3 (12.5%)	<0.0001 *
High	0	21 (87.5%)	

* statistically significant.

**Table 4 ijms-25-02561-t004:** Baseline characteristics of oropharyngeal cancer patients and control group.

		EBV				*p*	Total Patients		Control Group		*p*
		Positive		Negative							
		N	%	N	%		N = 110	%	N = 40	%	
Sex	Female	8	13.8	7	13.5	0.9999	15	13.8	6	15.0	0.7957
	Male	50	86.2	45	86.5		95	86.2	34	85.0	
Age	50–59	27	46.6	24	46.2	0.1116	59	53.4	21	52.5	0.9999
	60–79	31	53.4	28	53.8		51	46.6	19	47.5	
Place of residence	Urban	41	70.7	36	69.2	0.1667	77	70.7	28	70.0	0.9999
	Rural	17	29.3	16	30.8		33	29.3	12	30.0	
Smoking	≤10 * >10	28 10	48.3 17.2	25 10	48.1 19.2	0.8427	53 20	48.3 18.2	16 10	40.0 25.0	0.9999
	No	20	34.5	17	32.7		37	34.5	14	35.0	
Alcohol abuse	≤10 ** >10	18 10	31.1 17.2	15 10	28.8 19.3	0.9834	53	48.3	19	47.5	0.9999
	No	30	51.7	27	51.9		57	51.7	21	52.5	
G	G1	19	32.8	17	32.7	0.9997					
	G2	30	51.7	27	51.9						
	G3	9	15.5	8	15.4						
T	T1	7	12.1	8	15.4	0.9505					
	T2	27	46.6	22	42.3						
	T3	16	27.6	15	28.8						
	T4	8	13.7	7	12.1						
	N0	23	39.7	22	42.3						
N	N1	11	19.0	10	19.2	0.9844					
	N2	14	24.1	11	21.2						
	N3	10	17.2	9	17.3						
M	M0	58	100.0	52	100.0						

Pearson’s chi square Test; * packs/week; ** drink/week.

## Data Availability

Due to privacy and ethical concerns, the data used in this study are available from the corresponding author upon reasonable request.

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
