# Peer review of "Could MMP3 and MMP9 Serve as Biomarkers in EBV-Related Oropharyngeal Cancer"

_ijms, 2024, doi:10.3390/ijms25052561_

Round 1
Reviewer 1 Report
Comments and Suggestions for Authors
The topic is important and timely. The text reads well in general. I have minor suggestions. In the introduction, considering its incidence raise in the last years, advances in HPV-related disease management should be stressed. Therefore, text would be enhanced by addition of PMID: 31097964 and PMID: 33239190 references to better contextualize the issue at hand in oncologic scenario. It should be useful to provide smoke and alcohol details. Please add data (current smoker would have either ≤10 packs/years or >10 packs and alcohol consumer ≤ 21, and > 21 unit/week) at least in Table 4. Strengths of the study supporting evidence and validity of the research should be better reported. Please provide the complete form before providing the abbreviated form in the full-text (for instance, see line 73 and 76) and once provide the abbreviated form, use it in the full-text (for instance, see line 58 and 62). Figure 3 is not well-legible.
Author Response
Dear Reviewer,
We thank You for your comments on our paper. To guide the review process, we have copied the Reviewers comments in bold italics. Our responses are in regular font. We have responded to all the Referee comments and made alterations in the manuscript.
The topic is important and timely. The text reads well in general. I have minor suggestions.
- In the introduction, considering its incidence raise in the last years, advances in HPV-related disease management should be stressed. Therefore, text would be enhanced by addition of PMID: 31097964 and PMID:33239190 references to better contextualize the issue at hand in oncologic scenario.
In the introduction, taking into account the increase in the incidence of this disease in recent years, it was emphasized that, in accordance with the reviewer's request, both proposed literature items were cited in the introduction, which actually enriched the issues discussed in the oncological aspect.
- It should be useful to provide smoke and alcohol details. Please add data (current smoker would have either ≤10 packs/years or >10 packs and alcohol consumer ≤ 21, and > 21 unit/week) at least in Table 4. Strengths of the study supporting evidence and validity of the research should be better reported.
As suggested by the Reviewer, Table 4 provides details regarding smoke and alcohol, i.e. current smoker ≤10 packs/week or >10 packs/week and alcohol consumer ≤ 10 and >10 drinks/week). However, this aspect was not the focus of the current study and was therefore not analyzed.
- Please provide the complete form before providing the abbreviated form in the full-text (for instance, see line 73 and 76) and once provide the abbreviated form, use it in the full-text (for instance, see line 58 and 62).
I agree with the Reviewer. Therefore, the principle applied throughout the manuscript is to provide the full form of abbreviations used in the manuscript where they were used for the first time.
- Figure 3 is not well-legible.
Fig 3 has been enlarged a bit and I think it's legible now.
I would like to thank the Reviewer for your valuable advice and comments. I hope that the revised and slightly modified version of the manuscript will be more understandable to the reader. Best regardsMałgorzata Polz-Dacewicz MD PhD, Professor

Reviewer 2 Report
Comments and Suggestions for Authors
Research manuscript entitled, “Could MMP3 and MMP9 serve as biomarkers in EBV-related oropharyngeal cancer” related to the head and neck cancers is an interesting work establishing the involved mechanism of MMP3 and MMP9 as a biomarker in the cancer disease. The work has been clearly described and correlated with the obtained outcomes to hypothesize the involved mechanism. Still I’m having some suggestions to improve the readership of the manuscript:
1. Full form of the abbreviations used in the manuscript should be provided where they were used firstly for the better understanding of the audience and their use should be avoided in the abstract. Eg. EBV, OPSCC, etc.
2. The authors have mentioned that MMPs are regenerative in normal physiological condition by contributing for remodelling, embryonic development, wound healing, etc. but also in pathological conditions like inflammation, carcinogenesis and migration they are destructive in nature by degrading many components in extracellular matrix, the involved mechanism in the said process should be explained for better understanding of the statement.
3. The authors have stated, “Many MMPs have been shown to be overexpressed in head and neck cancer MMP-1, MMP-2, MMP-3, MMP-7, MMP-8, MMP-9, MMP-10, MMP-11, MMP-13, and MMP-14”, but selected only MMP3 and MMP9 for their research. What is the criteria for selection of these 2 MMPs as biomarkers for HNC and ignoring the others. It should be justified.
4. Authors are using the statistical data related to cancer from 2020, why they are not considering some recently updated data from 2023 or afterwards. It should be updated for better view of the disease impact on global human population.
5. Repetition of statistical data should not be done in conclusion section of the manuscript.
6. Conclusion should be rephrased to highlight the observed outcomes of the study and concluding the generated hypothesis from the same with some impactful evidences for the proposed hypothesis.
These points should be resolved for betterment of the manuscript and make it suitable for enhanced readership.
Author Response
Dear Reviewer,
We thank You for your comments on our paper. To guide the review process, we have copied the Reviewers comments in bold italics. Our responses are in regular font. We have responded to all the Referee comments and made alterations in the manuscript.
Research manuscript entitled, “Could MMP3 and MMP9 serve as biomarkers in EBV-related oropharyngeal cancer” related to the head and neck cancers is an interesting work establishing the involved mechanism of MMP3 and MMP9 as a biomarker in the cancer disease. The work has been clearly described and correlated with the obtained outcomes to hypothesize the involved mechanism. Still I’m having some suggestions to improve the readership of the manuscript:
- 1.Full form of the abbreviations used in the manuscript should be provided where they were used firstly for the better understanding of the audience and their use should be avoided in the abstract. Eg. EBV, OPSCC, etc.
I agree with the Reviewer. Therefore, the principle applied throughout the manuscript is to provide the full form of abbreviations used in the manuscript where they were used for the first time.
- The authors have mentioned that MMPs are regenerative in normal physiological condition by contributing for remodelling, embryonic development, wound healing, etc. but also in pathological conditions like inflammation, carcinogenesis and migration they are destructive in nature by degrading many components in extracellular matrix, the involved mechanism in the said process should be explained for better understanding of the statement.
This paragraph, in line with the Reviewer's suggestion, has been expanded.
- 3.The authors have stated, “Many MMPs have been shown to be overexpressed in head and neck cancer MMP-1, MMP-2, MMP-3, MMP-7, MMP-8, MMP-9, MMP-10, MMP-11, MMP-13, and MMP-14”, but selected only MMP3 and MMP9 for their research. What is the criteria for selection of these 2 MMPs as biomarkers for HNC and ignoring the others. It should be justified.
There would not be enough clinical material to test a panel of metalloproteinases - This was added to the limitations of the study.
However, the main reason for this choice was the inspiration from the research of other authors. We were inspired to choose these MMPs by two publications in particular. The first one by Li et al. in which the authors suggests that the combination of MMP3 activity and EBV antibodies may be a useful biomarker in the diagnosis of NPC. In turn, in the second publication by Lan et al. discovered that, in addition to MMP 9, the new target gene is MMP 3, the expression of which is increased by Zta - a lytic transactivator of Epstein-Barr virus (EBV). Both researchers focused on NPC. However, we wanted to test whether these MMPs might play a similar role in EBV-related oropharyngeal cancer.
This fragment was added to the Introduction where the purpose of the study was presented.
- 4.Authors are using the statistical data related to cancer from 2020, why they are not considering some recently updated data from 2023 or afterwards. It should be updated for better view of the disease impact on global human population.
This fragment has been corrected. The latest data available in Globocan concerns the year 2022. And such data was quoted. More recent epidemiological data for Poland are not available.
- 5.Repetition of statistical data should not be done in conclusion section of the manuscript.
In line with the Reviewer's comment, this has been corrected.
- Conclusion should be rephrased to highlight the observed outcomes of the study and concluding the generated hypothesis from the same with some impactful evidences for the proposed hypothesis.
As suggested by the Reviewer the conclusions have been redrafted to emphasize the observed study results.
I would like to thank the Reviewer for your valuable advice and comments. I hope that the revised and slightly modified version of the manuscript will be more understandable to the reader.
Best regards
Małgorzata Polz-Dacewicz MD PhD, Professor
.

Reviewer 3 Report
Comments and Suggestions for Authors
Rationale for the study has a high level of significance because there is growing evidence that microbiome (e.g., bacteria, virus, fungi) play a significant roles in the initiation and promotion of oropharyngeal carcinoma.
Furthermore, identification of a herpesvirus increase incidence and associated for monitoring OPC.
There is a paucity of publications that studies the microbiome related to the presentation of oropharyngeal carcinoma (OPC) and fewer reports that identify the presence of herpesviruses, such as EBV. The relationship between ENV and presence of OPC is characterized by clinical criteria (e.g., grade, stage, nodal involvement, etc..). The results from the serum are sufficient to support the presence of EBV and enhance expression of MMP 3/. These MMPs are reported toe be important for modifying the extracellular matrix and mesenchymal tissues for metastatic expansion of local growth. The additional identification of anti-EBV antibodies adds to the rigor and reproducibility of the findings.
However the discussion need improvement to place into context the findings.
Here are some suggestions:
EBVs are often identified with expression of MMP3/9 in person with patient examined for periodontal disease and have long COVID-19. This shows a significant alteration in oral microbiome will heighten the risk for persistent infection and depressed oral mucosal and systemic immunity
MMPs (e.3/9) are identified with poor oral health because they are produced by oral pathogens. This suggests prior poor-oral health (e.g.,, tobacco product or alcohol use) may pre-condition oral-pharyngeal tissues to permit infection and enhanced risk for reactivation of previous dormant /latent EBV. Notable is an overlapping of adhesion molecules between the oral bacterial pathogens, both extracellular and intracellular, to EBV (e.g., heparan sulfate, lectins-glycoproteins). Importantly the MMPs usually are released as a shower that target CDs on the host epithelial and immune cells. Upon viral adherence there is a triggering of host pathogen pattern recognition response (PRR: DAMP, PAMP, AMP), at the cell membrane and endosomal lysosomal membrane followed by a signal (e.g., STASTs-NFATS) that resolves into a E3 ubiquitin ligase and/or NF-KB transcriptional expression. This activity elicits cytokine host responses that include release of immunosuppressive activity blocking cytotoxicity and suppressing tumor immune surveillance. Furthermore, identification of EBV and related microbiome dysbiosis release metabolites (e.g., essential and non-essential amino acids) and endopeptidases (e.g., cysteine/serine proteases, trypsin--like enzymes, serine endopeptidases). Taken together it is the microenvironment interacting with the host oral-pharyngeal tissues producing host DNA and RNA instability and loss of oral mucosal tumor immune surveillance that enhances risk for OPC and identifies the importance of the MMP and EBV biomarker identification. Please note HPVs another dsDNA viral family that infects the OPC tissues are opportunistic viruses requiring a loss of epithelial lining integrity to integrate and contribute to E3 ubiquitin ligase-proteosome system regulation and loss of checkpoint control as prior infection by oral pathogens and EBV has damaged the oral mucosal lining.
Author Response
Dear Reviewer,
We thank You for your comments on our paper. To guide the review process, we have copied the Reviewers comments in bold italics. Our responses are in regular font. We have responded to all the Referee comments and made alterations in the manuscript.
Rationale for the study has a high level of significance because there is growing evidence that microbiome (e.g., bacteria, virus, fungi) play a significant roles in the initiation and promotion of oropharyngeal carcinoma.
Furthermore, identification of a herpesvirus increase incidence and associated for monitoring OPC.
There is a paucity of publications that studies the microbiome related to the presentation of oropharyngeal carcinoma (OPC) and fewer reports that identify the presence of herpesviruses, such as EBV. The relationship between ENV and presence of OPC is characterized by clinical criteria (e.g., grade, stage, nodal involvement, etc..). The results from the serum are sufficient to support the presence of EBV and enhance expression of MMP 3/. These MMPs are reported toe be important for modifying the extracellular matrix and mesenchymal tissues for metastatic expansion of local growth. The additional identification of anti-EBV antibodies adds to the rigor and reproducibility of the findings.
However the discussion need improvement to place into context the findings.
Here are some suggestions:
EBVs are often identified with expression of MMP3/9 in person with patient examined for periodontal disease and have long COVID-19. This shows a significant alteration in oral microbiome will heighten the risk for persistent infection and depressed oral mucosal and systemic immunity
MMPs (e.3/9) are identified with poor oral health because they are produced by oral pathogens. This suggests prior poor-oral health (e.g.,, tobacco product or alcohol use) may pre-condition oral-pharyngeal tissues to permit infection and enhanced risk for reactivation of previous dormant /latent EBV. Notable is an overlapping of adhesion molecules between the oral bacterial pathogens, both extracellular and intracellular, to EBV (e.g., heparan sulfate, lectins-glycoproteins). Importantly the MMPs usually are released as a shower that target CDs on the host epithelial and immune cells. Upon viral adherence there is a triggering of host pathogen pattern recognition response (PRR: DAMP, PAMP, AMP), at the cell membrane and endosomal lysosomal membrane followed by a signal (e.g., STASTs-NFATS) that resolves into a E3 ubiquitin ligase and/or NF-KB transcriptional expression. This activity elicits cytokine host responses that include release of immunosuppressive activity blocking cytotoxicity and suppressing tumor immune surveillance. Furthermore, identification of EBV and related microbiome dysbiosis release metabolites (e.g., essential and non-essential amino acids) and endopeptidases (e.g., cysteine/serine proteases, trypsin--like enzymes, serine endopeptidases). Taken together it is the microenvironment interacting with the host oral-pharyngeal tissues producing host DNA and RNA instability and loss of oral mucosal tumor immune surveillance that enhances risk for OPC and identifies the importance of the MMP and EBV biomarker identification. Please note HPVs another dsDNA viral family that infects the OPC tissues are opportunistic viruses requiring a loss of epithelial lining integrity to integrate and contribute to E3 ubiquitin ligase-proteosome system regulation and loss of checkpoint control as prior infection by oral pathogens and EBV has damaged the oral mucosal lining.
As per Your suggestion, a paragraph has been added at the end of the discussion to briefly provide a broader context of the processes occurring in the TME and the role of the complex microbiome, of which EBV is one of the components. However, the aim of this study was not to analyze environmental factors (smoking and alcohol consumption). Other aspects are beyond the scope of this article. However, they will be taken into account in future, differently planned studies.
The association between EBV and the development of nasopharyngeal cancer is well documented. We have been conducting research on EBV related HNC for several years. In one of our team's first studies on the prevalence of oncogenic viruses in oropharyngeal cancer, EBV was detected in 53.3% of cases. Since then, we have been conducting research on various aspects of EBV-related oropharyngeal cancer. We tested various biomarkers in serum, saliva and tumor tissue.
In our recent study (Cancers 2024), we examined the serum prevalence and the level of anti-Zta and anti-LMP1 antibodies. We concluded that combined antibody testing should be performed to increase diagnostic accuracy. In the same group of patients, mentioned above, the levels of MMP 3 and MMP 9 were analyzed to determine their usefulness as diagnostic and prognostic markers in EBV positive OPSCC patients.
I would like to thank the Reviewer for your valuable advice and comments. I hope that the revised and slightly modified version of the manuscript will be more understandable to the reader.
Best regards
Małgorzata Polz-Dacewicz MD PhD, Professor
